# The Glycocalyx: The Importance of Sugar Coating the Blood-Brain Barrier

**DOI:** 10.3390/ijms25158404

**Published:** 2024-08-01

**Authors:** Candis Dancy, Kaitlyn E. Heintzelman, Moriah E. Katt

**Affiliations:** 1Department of Chemical and Biomedical Engineering, West Virginia University, Morgantown, WV 26506, USA; crd0036@mix.wvu.edu (C.D.); keh00023@mix.wvu.edu (K.E.H.); 2School of Medicine, West Virginia University, Morgantown, WV 26506, USA; 3Department of Neuroscience, School of Medicine, West Virginia University Health Science Center, Morgantown, WV 26506, USA

**Keywords:** glycocalyx, blood–brain barrier, drug delivery, BBB disruption, ischemic stroke, neuroinflammation

## Abstract

The endothelial glycocalyx (GCX), located on the luminal surface of vascular endothelial cells, is composed of glycoproteins, proteoglycans, and glycosaminoglycans. It plays a pivotal role in maintaining blood–brain barrier (BBB) integrity and vascular health within the central nervous system (CNS), influencing critical processes such as blood flow regulation, inflammation modulation, and vascular permeability. While the GCX is ubiquitously expressed on the surface of every cell in the body, the GCX at the BBB is highly specialized, with a distinct composition of glycans, physical structure, and surface charge when compared to GCX elsewhere in the body. There is evidence that the GCX at the BBB is disrupted and partially shed in many diseases that impact the CNS. Despite this, the GCX has yet to be a major focus of therapeutic targeting for CNS diseases. This review examines diverse model systems used in cerebrovascular GCX-related research, emphasizing the importance of selecting appropriate models to ensure clinical relevance and translational potential. This review aims to highlight the importance of the GCX in disease and how targeting the GCX at the BBB specifically may be an effective approach for brain specific targeting for therapeutics.

## 1. Introduction

The endothelial glycocalyx (GCX) is a mesh, gel-like layer of polysaccharides that extends out from the cell membrane into the vascular lumen. This structure is important for the integrity of the cell structure, as well as for communication between other cells and maintaining cellular homeostasis [1]. Acting as the first physical barrier of protection for cells against foreign particles, the GCX is especially critical in the central nervous system (CNS), specifically at the blood–brain barrier (BBB). The BBB separates the CNS from the peripheral circulation, requiring specialized functions to be carried out by the endothelial GCX, such as the identification of pathogens and subsequent communication with immune cells [2]. Recent technical advances have enabled further GCX research, revealing the dynamic nature of the GCX and its involvement in various diseases [3]. Moreover, the recognition of the GCX as a pivotal factor in cellular homeostasis has led to increased exploration into using the GCX as a therapeutic target for disease [4]. As research continues to unveil the complexities of the GCX, there is great promise in contributing to the development of innovative and targeted therapeutic strategies utilizing the GCX.

The GCX functions as a protective and interactive layer of glycoproteins, proteoglycans, and glycosaminoglycans on the cellular surface, composed predominantly of heparan sulfate, chondroitin sulfate, hyaluronic acid, syndecans, and glypicans [1]. The GCX houses proteins in the cell adhesion molecule family, such as the intercellular- (ICAM), platelet endothelial- (PECAM), and vascular cell- (VCAM) adhesion molecules, shielding them from casual interaction [5]. For a comprehensive review of the endothelial glycocalyx, see the recent publication by Foote et al. [1]. This review will focus specifically on the role and importance of the GCX at the BBB.

The composition of the GCX is adaptable to the functional needs of its anatomical location [6]. Within the CNS, the GCX helps to form and maintain the BBB, and much like the brain endothelial cells that line the cerebrovasculature, it has specialized properties that enhance this protection. Exhibiting a negative charge throughout all cell types, the negative surface charge density is much higher at the GCX within the BBB compared to other locations [7]. Therefore, the GCX composition and structure in the brain may be different compared to the periphery (Figure 1).

The role of certain GCX components in BBB function is still unclear and requires deeper understanding. Despite the complex problem existing with pursuing accurate identification of the GCX, tools are currently in development to mitigate this issue. Recent studies have developed protocols that facilitate the characterization of GCX components, which can aid in the development of this understanding. Specifically, the use of liquid chromatography coupled with mass spectrometry (LC-MS) has shown success in facilitating GCX characterization [11]. Recently, studies have begun looking into novel ways to characterize the specific structural features of GCX components. For instance, studies have shown success employing lamprey-derived smart anti-glycan reagents (SAGRs) with unique amino acid sequences and diverse binding patterns to gain insight into glycan expression within the GCX with a specific focus on their structural arrangements [12]. Utilizing the information gained from this characterization can assist in the selection of targets for therapeutic development.

Beyond the intricacies of its composition and structural features, the GCX is presented as a potential target for therapeutic interventions. The GCX at the BBB is especially promising for targeting CNS diseases, given its pivotal roles in regulating molecular transport and maintaining barrier integrity. Applying lessons that scientists have learned from vascular GCX outside of the CNS gives valuable insight into how GCX may function in disease at the BBB where fewer studies are available. Exploitation of the GCX offers a strategic pathway toward influencing BBB permeability and selectivity, providing opportunities to target neurological disorders that are often significant challenges in drug delivery. This review sets out to explore the therapeutic potential inherent in targeting the GCX and offer insights into the latest developments and strategies for advancing therapeutic interventions in CNS disorders utilizing the GCX at the BBB.

## 2. Endothelial Glycocalyx in the Brain

The BBB is a highly specialized and selective barrier that separates the circulating blood from the brain parenchyma. It is formed by a complex network of endothelial cells, astrocytes, and pericytes that line the blood vessels of the CNS [13]. Endothelial cells compose the physical barrier, forming tight junctions that effectively eliminate paracellular transport, necessitating the high abundance of selective transport systems to control and facilitate the passage of molecules into the brain [14,15]. For molecules that are unable to gain passage across the BBB by passive diffusion, some of these transport mechanisms include carrier-mediated transport, receptor-mediated transport, and adsorptive transcytosis [16]. While the BBB is crucial for maintaining a stable and protected environment for the brain, safe from pathogens, toxins, or other stimuli that could damage this key organ, it also serves as a therapeutic barrier [17,18].

### 2.1. Glycocalyx Composition

Several studies have shown that particular components of the GCX are expressed at higher levels in the brain GCX, including chondroitin sulfate, heparan sulfate, phosphatidylinositol, and phosphatidylserine (Figure 1b) [8,9,10]. Beyond the distinct expression profiles of these GCX components, research by Suzuki et al. has unveiled varying side chain expressions on glycoproteins within the brain’s GCX compared to other body locations [19]. Employing lectin staining, Suzuki et al. demonstrated that Concanavalin A (ConA), a glycoprotein mannose side chain binder, exclusively bound to the murine brain endothelial GCX, while *Dolichos biflorus* agglutinin (DBA), a *N*-acetylgalactosamine binder, exhibited binding to the endothelial GCX of both the murine brain and lung capillaries [19]. These findings suggest differences in glycoprotein expression in the GCX at the brain compared to other organs. However, it is important to note that the precise composition of the GCX remains unknown and is potentially highly variable across individuals, influenced by factors such as age and health conditions like obesity [20].

Additionally, it has been found that the GCX in the lumen of the cerebral capillaries covers a significantly larger surface area than that found in other capillaries, such as the cardiac and pulmonary capillaries [21]. Ando et al. used transmission electron microscopy (TEM) to demonstrate that in mouse cerebral capillaries, 40.1 ± 4.5% of the endothelial surface was covered by GCX, contrasting with the 15.1 ± 3.7% observed on the endothelial surface of the heart capillaries and 3.7 ± 0.3% observed on the endothelial surface of the lung capillaries [21]. Additionally, the average thickness of the endothelial GCX in the brain was measured at 301.0 ± 111.8 nm, surpassing the average thickness of that found in the heart, 135.5 ± 59.7 nm, and lungs, 65.4 ± 28.4 nm [21]. It is important to note that GCX thickness and coverage in this study was observed using TEM, and variation sample preparations may result in different observed thicknesses; however, the trend remains consistent.

### 2.2. Physical Barrier

The GCX forms an intricate, gel-like mesh layer of polysaccharides that extends outward from the cell membrane into the vascular lumen, thus constituting a robust physical barrier. Kutuzov et al. used two-photon microscopy to uncover that the brain’s GCX mainly restricts large molecules, allowing small molecules such as sodium fluorescein (376 Da) and Alexa Fluor (643 Da) to penetrate most of the GCX volume, while the larger 40 kDa and 150 kDa dextrans penetrated less than 60%, preventing them from interacting directly with the cell surface or junctions [22].

The function of the GCX is not limited to restricting the passage of small molecules but includes impeding bacteria, viruses, or associated toxins from infiltrating the cell [23,24]. In its intact state, the GCX acts as a safeguard, preventing the interaction of viral spike proteins (S-proteins) with their corresponding receptors on endothelial cells angiotensin-converting enzyme 2 (ACE2) [25]. This protective function is achieved through the GCX’s role as a structural barrier and its active involvement in the regulation of ACE2, the SARS-CoV-2 binding receptor expression, which is crucial for viral entry mechanisms [25]. However, in situations where the GCX is disrupted or reduced, the endothelial receptors become exposed, increasing the susceptibility of these endothelial cells to infection [25]. This implies that previous injury or disease may increase the risk of brain penetrating infection, consistent with clinical findings [26,27]. Studies have found an exacerbation of symptoms was observed in patients with multiple sclerosis (MS) following infection of SARS-CoV-2 compared to pre-existing symptoms [28].

The negatively charged nature of the GCX plays a pivotal role in maintaining the proper functioning of the BBB [7]. The GCX is composed of a magnitude of different membrane-bound negatively charged proteoglycans, glycosaminoglycans, glycolipids, and glycoproteins [3,6]. The BBB’s GCX is more negatively charged than that of endothelial cells elsewhere in the body due to the comparatively higher content of phosphatidylinositol and phosphatidylserine [10]. Specifically, bovine brain capillary endothelial cells exhibited a zeta potential of −15.28 ± 0.58 mV, while human red blood cells (RBCs) and platelets displayed similar zeta-potential values to human umbilical vascular endothelial cells at values of −10.80 ± 1.63 mV and −10.75 ± 1.17 mV, respectively [10]. The negative charge assists in the creation of an electrostatic barrier, which in turn helps to repel and hinder the passage of negatively charged molecules, pathogens, and toxins. Additionally, the negatively charged GCX serves as a repulsive barrier, preventing the entry of red blood cells and weakening the interactions of white blood cells (e.g., leukocytes) and platelets with the vessel walls [29]. This repulsion is vital in the maintenance of the orderly flow of RBCs within narrow capillaries while also controlling platelet and leukocyte adhesion, imparting a degree of immune privilege [30,31]. The heightened negative charge of the brain endothelial cells (BECs) increases this immune privilege above other vascular beds.

### 2.3. Mechanosensing

Beyond its role as a physical and charge barrier, the GCX also assumes the function of a mechanosensor to take in information regarding the shear forces induced by blood flowing within the capillary lumen. These mechanosensors discern the dynamic mechanical cues of the GCX’s local microenvironment. Notably, the endothelial GCX acts as a mechanosensor that modulates vascular response to mechanical forces such as shear stress through biochemical pathways to facilitate the maintenance of vascular tone and prevent coagulation, thereby facilitating the regulation of cerebral blood flow [32]. Previous studies have suggested that the hyaluronic acid glycosaminoglycans (GAGs) within the GCX may be what is functioning as the mechanosensor that mediates shear-induced nitric oxide (NO) production [33,34]. The information gathered from these mechanosensors triggers a cascade of signaling pathways and subsequent gene expression activation [34]. A well-documented example of this can be shown in the difficulty of generating in vitro models that are able to recapitulate the complexity and density of the GCX in vivo. For example, past studies have found that experiencing increased fluid shear stress stimulates endothelial cells to incorporate larger amounts of hyaluronan, specifically glucosamine-containing GAGs, into their respective GCX [35]. A recent study demonstrated that exposing the cerebrovascular GCX to a high-intensity blast induces time-dependent alterations to the GCX, ultimately leading to the upregulation of components such as heparan sulfate, heparan sulfate proteoglycan, and chondroitin sulfate [32]. Thus, the GCX’s unique role as a mechanosensor facilitates a local environment-specific modulation of endothelial morphology and function [36].

This has particular relevance in the BBB, as it has been shown that shear stress results in a significant upregulation of GCX core proteins and galectins [36]. Santa-Maria et al. showed the effects of fluid flow on gene expression, utilizing a massive analysis of complementary DNA ends sequencing (MACE-seq) to determine the changes in the regulation of endothelial, BBB, and GCX-related genes, as well as surface charge. More specifically, utilizing a microfluidic lab-on-chip (LOC) device to simulate fluid flow conditions, they investigated the impact of shear-induced gene expression in a BBB model [36]. Through subsequent MACE-seq analysis, Santa-Maria et al. identified specific genes and pathways responsive to shear stress within the BBB endothelial cells. Notably, genes involved in the formation of tight junctions, such as claudins and occludins, as well as transporters responsible for regulating BBB molecular traffic, showed significant expression level alterations in response to the shear stress [36]. This study thus sheds light on the intricate interplay between shear stress and GCX dynamics within the BBB microenvironment, suggesting the key role that shear-induced changes in GCX composition play in BBB function and regulation. 

Previous research indicates that exposure to shear stress enhances the expression of several endothelial GCX components, including heparan sulfate, chondroitin sulfate, glypican-1, and syndecan-1 [37]. Thus, exposure to shear stress facilitates the maintenance of the more negatively charged surface and denser endothelial GCX structure that is characteristic of the BBB’s GCX [36,38,39]. Prior studies have explored the mechanisms through which the shear stress-mediated changes in protein and gene expression occur. Notably, Wang et al. demonstrated how shear stress modulation of the Krüppel-like factor 2 (KLF2) pathway leads to increased synthesis and membrane localization of the HA-producing enzyme Hyaluronan synthase 2 (HAS2), resulting in a thicker glycocalyx layer [40]. These findings suggest that the presence of shear stress may play a critical role in the maintenance and use of in vitro models of the BBB and that the evaluation of potential brain trafficking or luminal binding of therapeutics may be hampered by the use of static cultures.

### 2.4. Vascular Permeability

Several studies have demonstrated that the GCX plays a significant role in regulating vascular permeability and inflammation, especially in the context of the BBB [41,42,43,44,45]. The shedding of hyaluronan and syndecan-1 following ischemic stroke in a rodent t-MCAO model was correlated with increased caveolae-mediated endocytosis across the BBB [42]. This GCX degradation resulted in an enhanced interaction between syndecan-1 and sarcoma (proto-oncogene) kinase (Src), thus facilitating a rapid modulation of the endothelial cell’s cytoskeletal proteins [42]. Alterations in the composition of the BBB’s GCX can also impact the regulation of vascular permeability. Hyaluronidase treatment in rats resulted in an observed increase in BBB permeability, assessed through Evans blue and IgG measurements [41]. The study by DeOre et al. further demonstrated the significance of GCX composition in governing BBB vascular permeability, demonstrating that the knockdown of CD44, a mechanosensitive hyaluronic acid binding protein within the GCX, led to an increased permeability of the BBB [43]. These findings underscore the critical involvement of the GCX in orchestrating vascular dynamics at the BBB, shedding light on the correlation between GCX disruption and BBB disruption. 

The GCX also plays a crucial role in immune regulation, functioning as both a robust physical barrier and a dynamic mediator through its involvement in immune system signaling. When GCX degradation occurs, its function in both capacities is impaired, resulting in the induction of a proinflammatory phenotype and increased leukocyte adhesion to the underlying endothelial cells [44,45,46]. Specifically, McDonald et al. found that GCX degradation disrupts a key negative feedback loop, resulting in reduced NO levels, increased NF-κB activity, and subsequent endothelial cell overstimulation characterized by elevated ICAM-1 expression and increased leukocyte adhesion, illustrating its role in immune regulation at the BBB [44]. Thus, the GCX acts not only as a physical barrier to the adhesion of immune cells but also plays a role in maintaining the non-reactive homeostatic phenotype of endothelial cells at the BBB, which is also a key determinant in leukocyte adhesion and subsequent infiltration.

The distinctiveness of the brain’s GCX highlights its significance in the intricate orchestration of the BBB dynamics. The insight provided by these studies has not only enhanced our understanding of the unique physiological environment of the brain and the role that the GCX plays at the BBB in health but also suggests that a healthy, intact GCX is required for normal BBB function. It also suggests that preventing GCX degradation in disease may be a potential therapeutic avenue for conditions such as stroke, MS, and AD (Alzheimer’s disease), where BBB integrity is compromised [42,47].

## 3. Disease and the Glycocalyx

While the GCX is essential for maintaining cellular homeostasis, alterations in GCX structure and function have been implicated in several diseases [48]. These alterations can influence cell adhesion, transport, and immune processes, which have significant implications for CNS health and the development of potential therapeutics. Much of our understanding of the role of GCX in disease comes from peripheral organs, with continually improving technology and improved disease models and attention to the GCX, we see an increase in knowledge about the role of the GCX in neurovascular health. However, much of our understanding of the importance and role of the GCX at the BBB have their foundation in the GCX of the periphery. As highlighted in Table 1, our understanding of the GCX at the BBB in CNS disorders is somewhat limited. These observations underscore the widespread impact of GCX alterations on different physiological processes in organ systems throughout the body, as well as highlight the mechanisms in immune response that are likely also observed within the CNS.

### 3.1. Peripheral Organ Systems

The degradation of the GCX has been observed in renal diseases such as chronic kidney disease (CKD), where vascular injury from fibrosis resulted in the upregulation of proteoglycans syndecan-1 and glypican-1 [49]. In psoriasis, an EC subset identified specifically to the disease actively responded to many cytokines, including IFN-γ, resulting in enhanced T cell adhesion and deterioration of the endothelial GCX that gave rise to the aggravation of skin inflammation [50].

Vascular integrity is compromised in diseases such as atherosclerosis due to disruptions associated with the endothelial GCX, leading to the infiltration of immune cells into the arterial intima [44]. This imbalance results in the accumulation of immune cells and cellular debris at the arterial wall, exacerbating the progression of the disease. Evidence from an atherosclerotic mouse model shows that inflammation facilitates GCX shedding, further promoting monocyte adhesion and macrophage infiltration [51]. Systemic inflammatory diseases can also have profound effects on the integrity of the GCX. The integrity of the GCX has been correlated to clinical outcomes in sepsis, where the increased thickness of the GCX in the early stage of the disease correlated to greater survival rates in later stages due to GCX degradation [52]. 

The GCX can also act as a stealth shield for cancer cells, shielding tumor cells from immune recognition. Abnormalities in the composition of the GCX have been observed in various cancer types, further impacting immune cell adhesion and migration, contributing to tumor invasion and metastasis [53]. Irregular vascular flow patterns common in cancer have been observed to contribute to flow-induced degradation of the GCX [54]. As described above, the GCX plays an important role in immune infiltration, and parallels have been drawn between immune cell rolling and adhesion with early steps in the extravasation cascade [55]. Thus, it is likely that GCX modification plays a similar role in metastasis. Understanding the involvement of the GCX in cancer progression can provide potential avenues for therapeutic intervention, including but not limited to brain metastasis.

A significant amount of research has been conducted to give more insight into the changes in the GCX caused by heightened immune responses presented within diseases such as malaria and coronavirus disease-2019 (COVID-19). Analysis of sugar epitopes for GCX components in postmortem cerebral malaria (CM) tissue confirmed increased shedding of the endothelial GCX in CM tissue, with decreases in *N*-acetyl glucosamine [GlcNac] and sialic acid residues, alongside an increase in the inflammatory marker ICAM-1 [56]. This correlation between the breakdown of the endothelial GCX and the elevation of biomarkers for inflammation in CM patients underscores the impact of GCX integrity and the inflammatory response. In COVID-19 patients, a fragmented vascular endothelial GCX was observed to be caused by the high affinity of severe acute respiratory syndrome coronavirus 2 (SARS-CoV-2) to the ACE2 receptor, which is noted to play an important role in inflammation [57]. This interaction leads to compromised barrier function of the endothelial GCX in COVID-19 patients, potentially contributing to greater disease severity. The paralleled disruptions observed in the endothelial GCX in CM and COVID-19 highlight the important role of the endothelial GCX in mediating immune responses, offering valuable insights for therapeutic strategies aimed at preserving GCX integrity in mitigating inflammation in these diseases.

These studies provide evidence that the GCX is a major factor in the efficiency of physiological processes outside of the CNS. Understanding the complex roles that the GCX plays within different organ systems is crucial for the development of targeted therapies to provide information into the relationship between GCX integrity and overall health, as many of these same functions are carried out by the GCX in the CNS, as seen with cerebral malaria.

### 3.2. Central Nervous System

A common hallmark of many CNS diseases, including ischemic stroke, is dysfunction of the BBB, and disruptions to the endothelial GCX are a major contributor to this dysfunction [58]. There is a positive correlation between the proteoglycan levels in plasma, an indicator of GCX shedding, and National Institutes of Health Stroke Scale (NIHSS) scores in individuals with minor stroke [59]. Syndecan-1 levels are being used to predict the response to tissue plasminogen activator (tPA) and mechanical thrombectomy in the treatment of ischemic stroke [60,61]. The increased level of this major GCX component in the bloodstream could serve as an important biomarker outside of ischemic stroke as an early predictor for disease severity and as a diagnostic tool [61]. Strategies aiming for the preservation of GCX integrity could serve to aid in the maintenance of BBB function by mitigating the impact of pathological processes in the brain [62].

**Table 1 ijms-25-08404-t001:** The potential role of GCX in CNS diseases including ischemic stroke, AD, MS, and DiGeorge syndrome.

Disease	Model System	Main Findings	Citation
Ischemic Stroke	Murine transient middle cerebral artery occlusion (t-MCAO)	GCX components hyaluronan and syndecan-1 display biphasic expression in recovery following stroke.	[42]
	t-MCAO	Decreased infarct size and inhibition of leakage following stroke achieved by inhibiting caveolae-mediated transcytosis at the BBB.	[63]
Alzheimer’s Disease	Humanized mouse/human amyloid precursor protein (APP)	Modification of the *N*-GCX component results in worse cognitive function in AD.	[64]
	APP mice	Positive correlation observed between syndecan-3 and amyloid plaque load in AD.	[65]
	Mutant human presenilin 1 (PS1) mice	Loss of endothelial GCX may be driven by enhanced neutrophil-vascular interactions in AD.	[66]
Multiple Sclerosis	Experimental autoimmune encephalitis (EAE)-induced C57BL/6 J mice	Increased presence of GCX degradation markers heparin sulfate, hyaluronan, and syndecan-1 in MS.	[67]
	EAE-induced C57BL/6 J mice	Proteoglycan binding reduces inflammation and inhibits remyelination in MS.	[68]
DiGeorge Syndrome	Human brain microvascular endothelial cells (HBMECs)	BBB permeability increases with decreasing trans-endothelial electrical resistance in DS.	[69]
	HBMECs	Heparan sulfate expression is disrupted in DS resulting in loss of tight junction at the endothelial GCX.	[70]

It is well known that during an ischemic stroke there is a biphasic opening of the BBB, and this relationship is maintained when looking at the GCX breakdown [42]. Initially, the breakdown of the BBB during stroke and the subsequent reperfusion injury exacerbates endothelial GCX degradation resulting in increased inflammation and oxidative stress within the affected brain tissue [58]. One recent study observed this biphasic change pattern in the endothelial GCX during the first week following t-MCAO in mice, which corresponded to the biphasic evolution of permeability at the BBB [42]. In this study, the first phase following stroke resembled the degradation of the GCX caused by reperfusion injury. In contrast, the second phase corresponded to a restorative process with a recovered thickness of the GCX, suggesting that the GCX at the BBB may transition from the dense representation in Figure 1b to more closely mimic the more sparse GCX seen elsewhere in the body represented in Figure 1a. Specifically, levels of hyaluronan and syndecan-1 in plasma peaked at 6 h following t-MCAO in mice, followed by a secondary peak after 7 days [42]. This observation acts as an indication of the potential of the GCX to reconstruct and repair itself following an ischemic event.

While these clear observations indicate that there is some change in the GCX following ischemic stroke, it is not entirely clear if this is entirely representative of the change in GCX composition in the brain, or rather, throughout the entire vascular system. A better understanding of this temporal relationship is needed. Understanding the dynamic nature of the endothelial GCX and the exploration of therapeutic strategies, such as biomimetic treatments, aimed at enhancing GCX repair and protecting BBB integrity after CNS injuries may be a valuable area for further study.

Dysfunction of the endothelial GCX at the BBB could be linked to the development of AD and other neurodegenerative diseases, as suggested by evidence of leakage at the BBB in AD patients accompanied by the secretion of proinflammatory cytokines and chemokines [65,71]. This now proinflammatory phenotype creates a positive feedback loop for extended tissue damage, as seen in many neurodegenerative diseases [72]. Damage of the endothelial GCX at the BBB caused by neuroinflammation was observed in patients with early-stage MS as indicated by the increased presence of GCX degradation markers heparin sulfate, hyaluronan, and syndecan-1 [68,73]. More research is required in this area to understand the extent of BBB dysfunction as caused by GCX degradation induced by neuroinflammation throughout the time course of the disease, as it is often difficult to separate causality and correlation in vivo.

GCX dysfunction has also been suggested to be involved in DiGeorge syndrome, (DS), a validated genetic risk factor for schizophrenia. The deletion or reduction in the Crk-like (CRKL) gene, which encodes for a crucial adapter protein responsible for forming tight junction skeletons, affects the BBB at the CNS through dysregulation of cell adhesion and junction stability [69]. Monocultured static-induced blood–brain barrier (iBBB) models derived from induced pluripotent stem cells of DS patients displayed an increased permeability of the BBB, GCX disruption, and observed deficits in the junction proteins [70]. It has been well established from this lab that the iBBB integrity of DS patients is compromised with greater trans-endothelial electrical resistance, increased permeability, and a lack of the endothelial GCX component heparan sulfate [70].

Much of our understanding of the GCX at the BBB is from increased soluble protein seen in plasma, relatively little is still known about the precise changes in the structure and function of the GCX in CNS diseases. From diseases impacting the vascular glycocalyx in peripheral tissues, we can gain an increased understanding, but there are still fundamental gaps in knowledge and methodology that will be able to answer many of these critical questions. The systemic implications of the GCX alterations, evident in both cancer progression and CNS diseases, further highlight the role of the GCX in health and disease. Understanding the temporal expression changes and shedding of the GCX during disease will further the understanding of its role in disease progression and pathology.

## 4. GCX in Model Systems

In the development of therapeutics targeting or interacting with the GCX, selecting an appropriate model that recapitulates its key features is vital. The GCX is essential for maintaining BBB integrity and overall vascular health, influencing processes such as blood flow, inflammation, and vascular permeability [74,75]. Thus, choosing the most accurate model is imperative in ensuring that research findings can effectively be translated into clinical applications.

A clear example of the importance of choosing the appropriate model can be seen in influenza. Spruit et al. have demonstrated that mice are not the ideal species to study influenza virus due to their glycan profile [76]. The influenza virus binds to sialic acid-containing glycans on the cell surface, specifically *N*-acetylneuraminic acid (Neu5Ac) and *N*-glycolylneuraminic acid (Neu5Gc) [76]. Mice are not naturally susceptible to the human influenza virus due to their predominant expression of α2,3-linked sialic acids with both Neu5Ac and Neu5Gc modifications, rather than the α2,6-linked Neu5Ac expressed in humans [76]. Ferrets have a more similar sialic acid content to that of humans and are more suitable for studying the influenza virus’s pathogenesis and evaluating potential treatments [76]. This example highlights why selecting the appropriate model for GCX-based research is critical, as such differences in GCX structure or function could potentially explain why some treatments fail in clinical trials despite success in preclinical studies.

Experimental models currently employed for studying the GCX fall into several key categories: animal models, in vitro cell culture models, ex vivo models, clinical advanced imaging techniques, and computational models [77]. Each category has its own strengths and limitations, and it is important that researchers choose the model that best aligns with their research focus. By doing so, researchers can increase the likelihood of their findings, leading to successful disease modeling, therapeutic development, and effective clinical interventions.

### 4.1. In Vivo Animal Models

Rodent models, particularly mice and rats, are extensively used in preclinical studies, offering vital insights into the effectiveness and safety of a multitude of therapeutic interventions. Beyond assessing these aspects, rodent models also yield valuable information about the systemic effects and potential off-target sites of drug accumulation. However, it is imperative to acknowledge a constraint in translating the findings from rodent models of the BBB and its associated GCX to clinical settings. Notably, the composition of the BBB’s GCX in rodents diverges from that in humans as a result of species-specific differences in gene expression [77,78,79]. For example, several studies have demonstrated that the human GCX differs from that of many other mammals, including rodents, due to the lack of the sialic acid *N*-glycolylneuraminic acid (Neu5Gc) and an increased abundance of the precursor *N*-acetylneuraminic acid (Neu5Ac), among other differences [76,80]. 

Studying disease-related changes in the GCX often involves the use of animal models to mimic human conditions. Table 2 summarizes key results from in vivo models of CNS related diseases. For example, in a study on ischemic stroke, a mouse model of transient middle cerebral artery occlusion (t-MCAO) revealed a biphasic pattern of endothelial GCX degradation and reconstruction, correlating with the timeline of BBB damage, increased endothelial transcytosis, and elevated plasma levels of syndecan-1, contributing to brain edema and neurological dysfunction [42]. By emphasizing the biphasic change in both the GCX and BBB integrity, this study provides insights that could inform therapeutic strategies aimed at enhancing GCX repair and protecting BBB integrity during the recovery phase following ischemic stroke [42]. Elevated plasma levels of syndecan-1 are now being used in the clinic as an indicator of patient prognosis following treatment [60,61], clearly indicating the value and importance of in vivo studies of GCX shedding. 

These studies are not limited to ischemic stroke; in vivo studies have shown the promise of the repair or attenuation of GCX damage in the treatment of a number of diseases that are known to impact the CNS. In studies on status epilepticus (SE) utilizing a mouse model, GCX degradation post-SE was observed alongside increased BBB permeability; however, heparin treatment mitigated GCX disruptions, leading to improved outcomes [81]. Research on fluid therapy in hemorrhagic shock using an acute hemorrhage murine model demonstrated that fluid resuscitation with hydroxyethyl starch (HES) solution protected the GCX, decreased vascular permeability, and significantly reduced plasma syndecan-1 levels, thereby improving survival rates [82]. In studies on cerebral malaria, *Plasmodium berghei* ANKA (PbA)-infected mice displayed severe depletion of endothelial GCX during the terminal phase of infection, which was correlated with increased plasma levels of sulfated GAGs and HA, serving as an early marker of endothelial cell activation and inflammation, facilitating leukocyte interactions and malaria-infected erythrocyte sequestration [84].

Researchers studying systemic and pulmonary inflammation in mice observed a reduction in thickness of the endovascular GCX, alongside increased syndecan-1, hyaluronic acid, and heparanase levels in the blood in cecal ligation and puncture (CLP) mice [83]. Rats treated with monocrotaline were used to observe the effects of pulmonary arterial hypertension (PAH) on the GCX to determine whether GCX destruction was involved in the development of PAH [85]. Additionally, in diabetes mellitus (DM), db/db mice models were utilized to identify injury to the endothelial GCX prior to the onset of endotoxemia in type 2 diabetes, in which outcomes were worsened in the disease with the extended migration of inflammatory cells that attenuated endothelial GCX synthesis [87]. These findings contribute significantly to understanding the relationship between GCX changes and the pathogenesis of various disorders that have a profound impact on the BBB. Studies using live imaging the human GCX at the BBB have shown preliminary success indicating a possibility that this may be a valuable tool in the future [86].

As a result of species-specific differences, rodent models should be viewed as ‘incomplete’ models, as while they are able to provide invaluable insight into some key processes, they are unable to capture all the physiological complexities involved in human health and disease [88]. However, they are still valuable models for understanding how changes in the GCX play a critical role in cerebrovascular health in disease.

### 4.2. In Vitro Models

Endothelial cell culture models of the GCX are invaluable tools for the preliminary screening of potential therapeutic targets while also providing researchers with a controlled environment to investigate the effect of these drugs or other chemical or biological stimuli on the BBB’s GCX [89]. However, despite their wide-arching benefits, these models have limitations in their ability to recapitulate the actual BBB and its associated GCX [90,91]. Previous studies have developed various in vitro models of the BBB aimed at enhancing barrier properties through the co-culturing of brain endothelial cells and pericytes or utilization of organotypic systems like microfluidic devices and organ-on-a-chip platforms [36,92,93,94,95]. These models offer a more physiologically relevant environment, structure, and functionality but face challenges such as complex culture requirements and size constraints [89]. Recent investigations have shown that the dynamic conditions captured in these models lead to a significant upregulation of GCX core proteins and galectins, which subsequently results in a more negatively charged surface and a denser endothelial GCX structure that more closely mirrors the physiological GCX [36]. Notably, in vitro models cultured under flow conditions exhibit a significantly thicker GCX compared to those that were cultured under static conditions [38,39]. In conclusion, the ongoing refinement of endothelial cell culture models and the exploration of innovative organotypic systems represent promising avenues for overcoming existing model limitations and deepening our understanding of the GCX. Table 3 summarizes key findings and advancements in modeling the BBB GCX in vitro. 

As recapitulating GCX composition in vitro is challenging, GCX editing has emerged as a potential strategy to improve GCX modeling. Diverse methods for GCX editing, such as small molecule inhibitors and analogs, synthetic glycopolymers, metabolic reprogramming with activated donor sugars, and CRISPR/Cas9-based pruning, have been identified [96]. One particularly promising technique of precision GCX editing that has been identified is the construction of de novo scaffolding. This method entails the use of synthetic glycoconjugates with adjustable architectures and functionalities, which consequently facilitates a high degree of control over the editing process [97,98]. Overall, this class of techniques enables researchers to further discern the specific contributions of individual GCX components and has a potential role in the development of in vitro models of the BBB’s GCX [97]. By meticulously controlling the composition and structure of the GCX in these models, researchers will be better able to replicate the environment of the BBB’s GCX in both health and disease states. This level of precision also has the potential to enable further exploration of how the GCX influences BBB permeability, immune regulation, and response to various pathological stimuli or states [96,99]. Consequently, GCX editing holds significant potential for advancing the characterization of the BBB’s GCX and contributing to the development of disease models as well as therapeutic testing. 

**Table 3 ijms-25-08404-t003:** Overview of in vitro GCX models and associated key findings.

Model Category	Model System	Main Advantages	References
Static Models	Static endothelial cell culture models	Provides a controlled environment for studying GCX components but lacks the complexity and physiological relevance of in vivo conditions.	[69,89,90,91]
	Co-culturing brain endothelial cells and pericytes	Enhances model relevance by better replicating the dynamic interactions and structure of the BBB, although some limitations remain.	[92,93]
Shear Flow Models	Endothelial cell cultures under flow conditions	Facilitates significant upregulation of GCX core proteins and galectins, resulting in a thicker and more physiologically accurate GCX structure compared to static cultures.	[36,37,38,40]
	Microfluidic devices	Provides dynamic conditions that closely mimic physiological flow and GCX structure, improving the accuracy of BBB and GCX studies despite fabrication and size challenges.	[94,95]
	Microfluidic devices with co-cultured cells	Enhances model relevance by replicating the dynamic interactions between brain endothelial cells and other cell types under shear flow conditions, thereby improving the accuracy of GCX studies.	[94,95]
GCX Editing	GCX editing with small molecule inhibitors and analogs	Allows precise manipulation of GCX components to investigate their role in BBB function and has potential therapeutic applications for many disease states, including DS.	[70,96]
	GCX editing with synthetic glycopolymers	Facilitates detailed control over GCX composition and structure, enhancing the ability to replicate in vivo conditions and study GCX-related processes.	[97,98,99]
	GCX editing with CRISPR/Cas9-based pruning.	Enables targeted exploration of specific GCX components, aiding understanding of their contributions to BBB permeability and immune responses.	[96]

The study of the BBB’s GCX is often neglected in vitro, as many models lack the effort and precision needed to accurately represent and characterize the GCX, with comparisons to in vivo data being almost non-existent. Advances in imaging preparation techniques have improved the in vitro study and characterization of the GCX. For example, Ebong et al. demonstrated that the use of rapid freezing/freeze substitution transmission electron microscopy (RF/FS-TEM) facilitated the in vitro stabilization of the GCX in its hydrated, protein-rich state [100]. Despite these advancements, current models still struggle to accurately replicate the BBB and its GCX, making it a difficult and neglected field of study [90,91].

### 4.3. Ex Vivo Models

Ex vivo models provide researchers with a deeper understanding of intricate biological processes outside of the living organism. This is especially significant in the setting of understanding human physiology and developing therapeutics, an area often constrained by the limitations of in vivo animal models [101,102,103]. For measurements and studies of the human GCX, scientists are largely limited to studying post-mortem tissue. Ex vivo models offer a more direct or species-relevant approach to studying the GCX [101,102]. Key findings from ex vivo models of the BBB GCX are summarized in Table 4. 

Imaging and assessment of the human cerebral microvasculature and its GCX have primarily been performed ex vivo through microscopic evaluation of structural changes, as well as vascular stainings and markers [86]. However, these ex vivo measurements often underestimate the actual thickness of the cerebrovascular GCX due to its collapse under such conditions, and functional analysis of the microvasculature cannot be performed ex vivo [86]. The variability of tissue handling post-mortem or in resected samples further complicates these measurements, potentially affecting protein content and, postulated similarly, altering glycan content critical for GCX function [104,105]. While specific studies on glycomics and glycans in post-mortem tissue are scarce, it is well established that different post-mortem intervals, storage conditions, and handling techniques can lead to degradation of proteins and mRNA, significantly impacting the result of proteomic and transcriptomic analyses [104,105]. Specifically, studies have shown that prolonged post-mortem intervals can result in substantial changes in protein expression levels and RNA integrity [104,105]. Therefore, the interpretation of ex vivo GCX measurements must consider these potential artifacts to ensure accurate representation of its in vivo condition.

This challenge in accurately measuring the GCX ex vivo underscores several limitations compared to in vivo evaluations. Traditional techniques such as TEM can enable visualization of the GCX but require extensive preparation that often causes the GCX to collapse, thus leading to inaccurate representations of its structure [100,106]. Techniques to mitigate this collapse, such as rapid freezing and freeze substitution techniques, have been explored and have shown some success, but they likely still lead to an underestimation of the true GCX thickness [100]. Other ex vivo microscopy methods, such as confocal and two-photon laser scanning microscopy, use fluorescent markers to analyze GCX components, facilitating both quantitative and qualitative analysis; however, these methods are still constrained by the GCX’s fragility in ex vivo conditions [107,108]. Recent advances in techniques such as stochastic optical reconstruction microscopy offer high-resolution insights but are similarly limited by the delicate nature of the GCX ex vivo [106]. As such, novel in vivo techniques have shown potential for more accurate GCX assessment without the issues introduced by ex vivo handling and preparation.

**Table 4 ijms-25-08404-t004:** Overview of ex vivo GCX models and associated key findings.

Characteristic Being Measured	Model System	Summary of Main Findings	References
GCX Thickness and Structure	Post-mortem tissue analysis with TEM	Provides structural insights but may underestimate GCX thickness due to collapse during preparation. Techniques such as rapid freezing have shown some success but still lead to underestimation.	[100,106]
	Post-mortem tissue analysis with stochastic optical reconstruction microscopy	Offers high-resolution insights but constrained by GCX delicacy in ex vivo conditions, leading to potential inaccuracies in thickness measurements.	[106]
	Post-mortem tissue analysis of human umbilical veins	Highlights discrepancies in GCX thickness between ex vivo and in vitro models, underscoring the need for accurate model validation.	[109]
Protein and RNA Integrity	Post-mortem tissue analysis for protein and RNA integrity	Prolonged post-mortem intervals and handling techniques cause protein and mRNA degradation, significantly impacting transcriptomic analyses.	[104,105]
GCX Component Analysis	Post-mortem tissue analysis with confocal and two-photon microscopy	Uses fluorescent markers to study GCX components but is limited by GCX fragility and potential artifacts from tissue handling and preparation.	[107,108]
	Post-mortem brain tissue analysis from children who died of CM	Analysis showed significant GCX shedding in CM, with decreased *N*-acetyl glucosamine and sialic acid residues. Elevated levels of inflammatory marker ICAM-1 highlights correlation between GCX breakdown and increased inflammation in CM patients.	[56]

Despite these challenges, ex vivo models have proven instrumental in unraveling the structure and thickness of the GCX in humans. For instance, Chappell et al. demonstrated a discrepancy in GCX thickness between their ex vivo model of human umbilical veins and their in vitro model of the same cell type [109]. Ex vivo models thus serve as tools for investigating physiological intricacies and a valuable reference point for validating the accuracy of in vitro models [110].

## 5. GCX Therapeutic Approaches

Therapeutic development for many CNS diseases is lagging compared to peripheral tissues [111,112]. There are many culprits to explain the high failure rate of CNS therapeutics, including lack of mechanistic disease understanding, lack of accurate models, and difficulty with delivering therapeutics specifically to the brain [112,113,114,115,116,117,118]. The GCX is emerging as a source for potential solutions [119]. Despite its complexity and the current gaps in understanding its role in various CNS diseases, the site-specific composition of the GCX has emerged as a promising target for CNS therapeutic development. To try to exploit this potential, three distinct approaches have been used and will be summarized here: utilizing the GCX to target therapeutics to the CNS, strategic modification of the host GCX, and disrupting the GCX for increased uptake (Figure 2).

### 5.1. Targeting Therapeutics to the CNS via the GCX

The role of the GCX in immune cell transport has been widely discussed here, but it also plays an important role in the transport of other molecules at the BBB. Brain vasculature is rich in transporters, facilitating the transport of many molecules into and out of the brain [13]. These transporters are housed within the GCX, and some may be tethered to the GCX, as VCAM and ICAM often are [74]. GCX-modulated receptor-mediated transcytosis is vital for maintaining the selective permeability of the BBB, promoting the controlled transport of molecules into the brain [42,126]. Receptor-mediated transcytosis is a highly studied method for improved CNS delivery, with the main transporters being studied being the insulin receptor, transferrin receptor, and CD98 [127,128]. However, these promising strategies have yet to identify a brain-specific receptor [129]. Recent studies have demonstrated notable success in the development of molecules targeted to specific GCX components, resulting in similar transport mechanisms with improved specificity [130].

Elevated levels of heparan sulfate within the cerebral capillary GCX have been utilized to enhance delivery to the brain. Joshi et al. demonstrated that neural stem cell (NSC) derived exosomes were able to efficiently cross an in vitro BBB without hampering the endothelial cell monolayer through interactions with the heparan sulfate proteoglycan (HSPG) component of the GCX and dynamin-dependent endocytosis [8]. Using an in vitro transwell model of the BBB, Joshi et al. investigated the specificity of the role of HSPGs in NSC-derived exosome uptake. They found that incubating cells with these NSC-derived exosomes in the presence of free heparin or Heparinase III (HSase) significantly reduced exosome uptake, indicating that HSPGs likely function as key mediators in this process [8]. However, whether HSPGs act as true internalization receptors for receptor-mediated transcytosis or merely as attachment sites for non-specific adsorptive endocytosis requires further investigation [8]. Furthermore, they demonstrated that the selected NSC-derived exosomes were able to transport protein cargo, a stand-in for potential therapeutic agents, across the in vitro BBB [8].

Exosomes have also been observed to bind effectively to glycoproteins containing sialic acid and *N*-acetyl-D-glucosamine at the BBB [131]. Subsequent administration of wheatgerm agglutinin (WGA), which is also bound to these glycoproteins, facilitated the modulation of adsorptive transcytosis of the identified exosomes across the BBB [131]. These findings underscore the potential of leveraging GCX-modulated transcytosis, particularly those involving GCX components abundant in the brain GCX for the development of innovative, specifically targeted therapeutic strategies.

Exosomes and nanocarriers are not the only strategies used for GCX binding. Geoghegan et al. incorporated GCX component binding activity into a viral vector, resulting in improved brain vascular tropism and transcytosis [9]. By adding chondroitin sulfate binding activity to their Adeno-associated virus vector (AAV-GBM), they achieved brain tropism and enhanced BBB transcytosis [9]. This strategic modification highlights the diverse therapeutic approaches possible for targeting the BBB’s GCX and the potential for increasing and improving the trafficking of diverse therapeutic cargo loads. Recent studies, such as that conducted by Merkel et al., have identified AAV vectors, specifically AAV2, capable of traversing the BBB through binding to HSPGs within the brain microvascular cell GCX [132]. This interaction facilitates cellular transduction by promoting the internalization of AAV2 capsids into endosomal-like structures, ultimately leading to their nuclear translocation and gene expression, thereby unlocking their therapeutic potential [132].

Screening methodologies also offer a viable avenue for pinpointing molecules that have targeted affinity for the BBB’s GCX and brain tropism. Lamprey-derived variable lymphocyte receptors (VLRs) are a potentially valuable tool for this endeavor as they show enhanced binding to glucans and also have the potential to be used therapeutically [12,133]. Screening of VLRs against mouse brain endothelial cells facilitated the identification of VLRs that bound to a ‘glyco-signature’ necessary for the specific targeting of the BBB’s GCX, as opposed to the more generic GCX of endothelial cells at large [134]. In essence, the application of screening methodologies, such as that illustrated by the VLRs study, provides a promising pathway for the identification of therapeutic agents that have tropism for the BBB’s GCX.

### 5.2. Strategic Modification of the Host GCX

The strategic modification of the GCX has emerged as a novel approach to optimize drug delivery and leverage the GCX as a therapeutic entity. A significant number of therapeutic monoclonal antibodies target components of the endothelial GCX, including glycoproteins, proteoglycans, specific carbohydrate structures, glycolipids, and sialic acid residues, among others [135,136,137]. Increasing the expression of these specific GCX components holds promise for enhancing the efficacy of such monoclonal antibodies. Moreover, strategic GCX modification has the potential to directly alter disease progression pathways, thus presenting as a promising avenue for standalone therapies [97,122,123,124]. 

As previously described, cancer cells often have a different GCX composition to protect them from the immune system. An example of a monoclonal antibody that has been developed to target a GCX component is an anti-GD2 mAb therapy (e.g., dinutuximab) that is used clinically to treat pediatric neuroblastoma (Figure 2b) [120]. Despite the upregulation of disialoganglioside GD2 on tumor cells, a large proportion of the patients in these trials still exhibited progressive disease. A recent study demonstrated that the expression of GD2 could be further enhanced with the addition of sialic acid analogs and HDAC (histone deacetylase) inhibitors (Figure 2b) [121]. This enhancement bolstered the targeting potential of the anti-GD2 mAb, exemplifying the possibility that strategic modifications to the GCX can be used to optimize therapeutic outcomes.

Modifying the GCX has the potential to go beyond simply enhancing the expression of therapeutic targets; it can also serve to augment key signaling pathways [97]. When pre-adipocyte GCX is modified with synthetic heparan sulfate, there is an increased capacity for glucose clearance independent of insulin secretion through the attenuation of Wnt signaling, underscoring its potential utility in the treatment of type 2 diabetes (Figure 2c) [122]. This highlights the nuanced impact of strategically altering the GCX in regard to both therapeutic target expression and key signaling pathways while also presenting opportunities for future advancements in therapeutic strategies.

Evidence supporting this approach includes findings that antithrombin administration prevents endothelial GCX shedding during ischemia and reperfusion, leading to sustained endothelial permeability, reduced post-ischemic coronary resistance, and mitigation of tissue edema (Figure 2d) [123]. This suggests that antithrombin not only has therapeutic potential in managing ischemia-reperfusion injury but also highlights the broader concept of GCX repair and maintenance as a viable therapeutic strategy. Several studies have identified other potential avenues for GCX repair and preservation. For instance, the administration of hydrocortisone and antithrombin has been shown to preserve the endothelial GCX against inflammatory degradation initiated by TNF-α, maintaining barrier function (Figure 2d) [124]. In a similar manner, administration of Etanercept, a TNF-α receptor analog, significantly reduced GCX shedding following endotoxin application [138]. Furthermore, research into protease inhibitors like doxycycline or zinc chelators has shown promise in inhibiting MMP activity, leading to reduced GCX shedding and leukocyte adhesion in response to inflammatory and ischemic conditions [139,140]. Additionally, volatile anesthetics like sevoflurane and isoflurane have been found to ameliorate endothelial GCX destruction induced by inflammatory responses [141,142]. These studies underscore the multifaceted approaches to preserving and repairing the endothelial GCX, highlighting their potential to mitigate vascular damage and sustain barrier function in the context of inflammatory and ischemic challenge, thereby offering a potential therapeutic strategy for clinical applications.

### 5.3. Disrupting the GCX for Enhanced Therapeutic Uptake

An extreme version of GCX modification is the disruption of the GCX. Studies have indicated that disrupting the GCX results in increased permeability and holds the potential for facilitating the trafficking of pharmacologic agents across the BBB, thus offering a promising solution for the issue of the BBB in the treatment of many CNS diseases [42]. Xia et al. demonstrated that transcranial direct current stimulation (tDCS) can transiently disrupt the GCX, leading to an augmented BBB permeability, especially for large or charged solutes (Figure 2e) [125]. Despite the considerable potential in augmenting BBB permeability through GCX disruption, it is crucial to acknowledge the risks associated with such interventions. Previous research has shown that glycocalyx disruption and degradation are associated with greater levels of brain injury coupled with an augmented permeability of the BBB, which has the potential to potentiate the subsequent development of brain edema [41]. Therefore, while GCX disruption may facilitate increased permeability and improved trafficking of pharmacologic agents across the BBB, consideration should be given regarding the risks associated with GCX disruption, though there is evidence that the GCX may recover.

Researchers have also been leveraging the damaged GCX to improve therapeutic delivery. In inflammation, when there is a degradation of the GCX, ICAM-1 is exposed as previously described. Hsu et al. demonstrated that targeting ICAM-1 not only facilitated efficient binding and uptake of NCs by endothelial cells but also enhanced their transport across the BBB through CAM-mediated transcytosis [143]. Additionally, targeting ICAM-1 has shown promise for the intracellular delivery of therapeutics to neurons [144]. Building on these findings, Manthe et al. found that the valency of the NCs significantly impacted transcytosis efficiency, with an intermediate valency being optimal for balancing binding and uptake with the efficient release at the basolateral side of the endothelial cells [145].

## 6. Conclusions

There is still a lot to learn in the area of glycomics and glycoproteomics. Glycan expression is much more fluid and can be impacted by everything from genetics to diet and environmental cues [146,147,148]. There are still large looming questions about individual-to-individual variation in the glycome. It is clear, however, that understanding the glycocalyx is critical for understanding CNS disease pathobiology as well as the transport of potential therapeutics. The lack of inclusion of the GCX as an important characteristic of BBB models is a critical failure, particularly where transport mechanisms are studied, as it is only after a molecule or particle has navigated the GCX that it has access to the proteins on the surface of the endothelial cells. Improved understanding of the dynamic changes in the GCX will lead to novel therapeutic strategies and a greater understanding of CNS diseases.

## Figures and Tables

**Figure 1 ijms-25-08404-f001:**
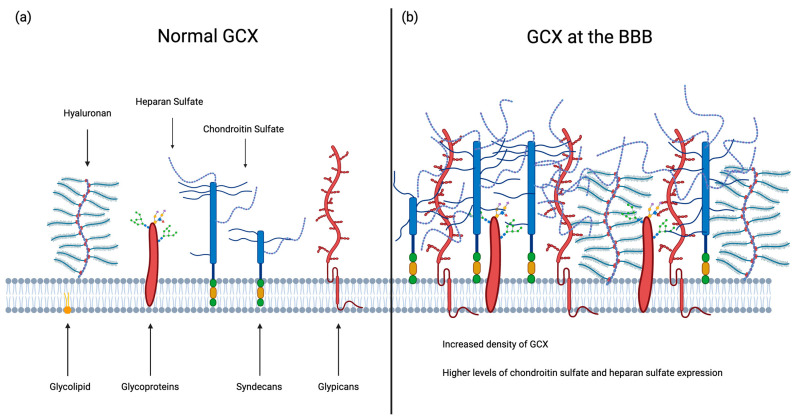
A representative comparison of the vascular glycocalyx in the periphery (**a**) and lining the interior cerebrovasculature at the BBB. (**a**) The mesh-layer of polysaccharides within a normal endothelial GCX includes, from left to right, glycolipids, hyaluronan with aggrecan containing brushes, glycoproteins, heparan sulfate (blue and purple chain) and chondroitin sulfate (solid blue chain) attached to syndecans, and glypicans [1]. (**b**) The endothelial GCX at the BBB contains the same major components as the peripheral GCX but exhibits a higher density, with notably higher levels of chondroitin sulfate and heparan sulfate expression; these function to stabilize and enmesh GCX components [8,9,10].

**Figure 2 ijms-25-08404-f002:**
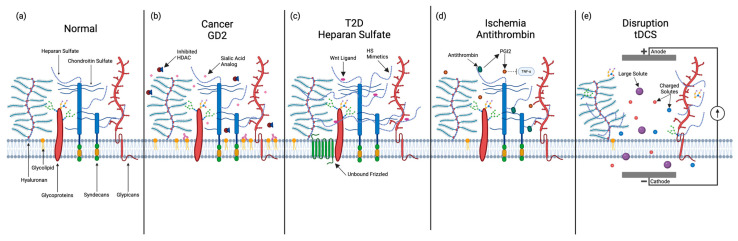
Potential avenues for therapeutic modulation of the GCX currently employed in preclinical studies. (**a**) The normal GCX includes hyaluronan, glycolipids, glycoproteins, heparan sulfate, chondroitin sulfate, syndecans, and glypicans [1]. (**b**) Sialic acid analogs and HDAC inhibitors enhance GCX GD2 expression in tumor cells. Improving anti-GD2 mAB targeting [120,121]. (**c**) Synthetic heparan sulfate modification of GCX attenuates Wnt signaling, boosting glucose clearance capacity [122]. (**d**) Antithrombin administration prevents endothelial GCX shedding during ischemia by stabilizing the GCX and promoting PGI2 release, inhibiting TNF-α-mediated degradation [123,124]. (**e**) Transient GCX disruption by tDCS increases BBB permeability, particularly for large or charged solutes [125].

**Table 2 ijms-25-08404-t002:** Overview of in vivo GCX disease models and associated key findings.

Measured Characteristic	Model System	Summary of Main Findings	References
Soluble GCX components in plasma	Mouse model of SE	GCX degradation occurs post-SE. Heparin treatment mitigated GCX disruptions, leading to improved outcomes by reducing BBB permeability and protecting GCX integrity.	[81]
	Acute hemorrhage murine model	Fluid resuscitation with HES solution protected the GCX, decreased vascular permeability, and reduced plasma syndecan-1 levels, thereby improving survival rates and outcomes in hemorrhagic shock.	[82]
	CLP mice for systemic and pulmonary inflammation	Observed reduction in thickness of endovascular GCX, and increased blood levels of syndecan-1, HA, and heparanase, indicating GCX shedding and degradation during systemic inflammation.	[83]
	PbA-infected mice for cerebral malaria	Severe endothelial GCX depletion during infection terminal phase correlated with increased plasma levels of sulfated GAGs and HA, serving as early marker of endothelial cell activation, inflammation, and facilitating leukocyte interactions.	[84]
	Human patients with acute ischemic stroke	Syndecan-1 levels in patient plasma can be used clinically as an indicator of patient prognosis following acute ischemic stroke treatment.	[60,61]
	C57BL/6 J mice and Lewis rats	Shedding of the GCX can serve as a biomarker for MS, with soluble, sugar-based GCX components being associated with disease severity.	[67]
GCX component expression	Monocrotaline-treated rats for PAH	GCX destruction observed in PAH development, suggesting GCX integrity is crucial for maintaining normal pulmonary arterial pressure and function.	[85]
	APPSWE-Tau transgenic mice	SDC3 expression on monocytes has a positive correlation with amyloid plaque load in the brain.	[65]
GCX thickness	Mouse model of t-MCAO	Observed biphasic pattern of endothelial GCX degradation and reconstruction, correlating with BBB damage, increased endothelial transcytosis, and elevated plasma syndecan-1 levels. These changes contribute to brain edema and neurological dysfunction.	[42]
	Human patients undergoing resective brain surgery	SDF imaging shows potential for in vivo assessment and functional analysis of the cerebral microcirculation and GCX.	[86]
BBB permeability	Rat model of t-MCAO	Storax treatment inhibited caveolae-mediated transcytosis at the BBB, reduced infarction size, and brain water content, with specific dose-dependent effects on protein expression.	[63]
Inflammatory cell migration	APP/PS1 mice	Endothelial GCX loss might be driven by enhanced neutrophil-vascular interactions in Alzheimer’s disease.	[66]
	Experimental Autoimmune Encephalomyelitis in mice	Surfen treatment reduced inflammation and immune cell infiltration in the CNS but inhibited remyelination by increasing CSPG expression.	[68]
	Db/db mice model for T2DM	Endothelial GCX injury was observed prior to endotoxemia onset, worsening outcomes due to extended inflammatory cell migration that attenuated GCX synthesis, indicating early GCX damage in diabetes progression.	[87]

## Data Availability

Individual data can be found in the referenced manuscripts.

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
