# Peer review of "The Glycocalyx: The Importance of Sugar Coating the Blood-Brain Barrier"

_ijms, 2024, doi:10.3390/ijms25158404_

Round 1

Reviewer 1 Report

Comments and Suggestions for Authors

Comments/suggestions

The endothelial glycocalyx is a network of membrane-bound proteoglycans and glycoproteins and influences the structure of the blood-brain barrier. GCX plays various physiological functions, including maintaining normal neuronal homeostasis therefore this is an important therapeutic target for CND diseases. The present article widely discusses GCX and examines the diverse model systems used in GCX-related research, emphasizing the importance of selecting appropriate models to ensure clinical relevance and translational potential.

Recommendation

1.      This article provides an in-depth knowledge of GCX-related models and therapeutics approaches.

2.      The research topic is interesting, well-planned, and has strong potential in CNS-related therapeutics.

3.      The article scientifically discussed each section and subsection with appropriate references.

4.      The article is error-free and has sufficient data to publish in a reputed journal like IJMS.

Minor suggestions

1.      Lines 308-312, provide a suitable reference for these lines.

2.      Lines 537-540, provide a suitable reference.

3.      If possible, provide a summary table for the in vivo, in vitro, and ex vivo models for GCX.

4.      If possible, provide a table for published articles showing the potential role of GCX in CNS diseases.

5.      Line 615 & 621, figure 3d it might be figure 2d.

6.      Line 621, figure 2d instead of figure 3d?

Author Response

Comments/suggestions

The endothelial glycocalyx is a network of membrane-bound proteoglycans and glycoproteins and influences the structure of the blood-brain barrier. GCX plays various physiological functions, including maintaining normal neuronal homeostasis therefore this is an important therapeutic target for CND diseases. The present article widely discusses GCX and examines the diverse model systems used in GCX-related research, emphasizing the importance of selecting appropriate models to ensure clinical relevance and translational potential.

Recommendation

  1. This article provides an in-depth knowledge of GCX-related models and therapeutics approaches.
  2. The research topic is interesting, well-planned, and has strong potential in CNS-related therapeutics.
  3. The article scientifically discussed each section and subsection with appropriate references.
  4. The article is error-free and has sufficient data to publish in a reputed journal like IJMS.

Response: The authors appreciate the reviewers kind depiction of the review. 

Minor suggestions 

  1. Lines 308-312, provide a suitable reference for these lines. 

Response: References supporting these statements have been added.

  1. Lines 537-540, provide a suitable reference. 

Response: A reference has been added supporting this information. 

  1. If possible, provide a summary table for the in vivo, in vitro, and ex vivo models for GCX. 

Response: The authors appreciate this excellent suggestion. Authors created summary tables for each of the GCX model types which are now found as tables 2-4.

  1. If possible, provide a table for published articles showing the potential role of GCX in CNS diseases.

Response: The authors appreciate this excellent suggestion, and this is now incorporated as table 1.

  1. Line 615 & 621, figure 3d it might be figure 2d. 

Response: Authors corrected mistype. 

  1. Line 621, figure 2d instead of figure 3d?

Response: Authors corrected mistype. 

Reviewer 2 Report

Comments and Suggestions for Authors

General comments:

Overall, this is a good and timely review of the glycocalyx of the BBB and its relevance to the diseases and potential therapeutics. However, several concerns need to be addressed. Please see below.

Specific comments:

1.     The title is missing leading since every mammalian cell has the surface glycocalyx. More accurately, it is suggested to be “The Glycocalyx: Sugar Coating of Blood Vasculature in the Central Nervous System”

2.     In Ln 42-44, it was said “The GCX also includes proteins in the cell adhesion molecule family, such as the intercellular- 43 (ICAM), platelet endothelial- (PECAM), and vascular cell- (VCAM) adhesion molecules”, not sure if this statement is correct. Cell adhesion molecules are not components of the GCX.

3.     More detailed captions should be given for Fig. 1.

4.     Ln 111-117, the length should be thickness. In addition, the observed thickness and coverage is from TEM. Sample preparations for different organs may have different artifacts for GCX coverage and thickness. This should be mentioned.

5.     It is suggested to have a list of acronyms for MACE-seq, LOC, HA, GAGs, KLF2, HAS2, t-MCAO, NO, Src, CM, etc.

6.     Be consistent with the words used such as “syndecan-1” and “Syndecan1”.

7.     There is no Fig. 3, “Fig. 3d” in Ln 615, etc, should be revised. Also other typos in the manuscript.

8.     Some recent articles were missing for the GCX of BBB, e.g., Li et al. 2024 (Cells 2024, 13, 190. https://doi.org/10.3390/cells13020190); Li et al, 2023 (Cells 2023, 12, 422. https://doi.org/10.3390/cells12030422); Xia et al, 2022 (APL Bioeng. 6, 016101, 2022 https://doi.org/10.1063/5.0064381); Li et al, 2021 (Cells 202110, 2576. https://doi.org/10.3390/cells10102576). It is suggested to perform more thorough literature review by using more combinations of key words.

Author Response

Reviewer 2

Overall, this is a good and timely review of the glycocalyx of the BBB and its relevance to the diseases and potential therapeutics. However, several concerns need to be addressed. Please see below.

Specific comments: 

  1. The title is missing leading since every mammalian cell has the surface glycocalyx. More accurately, it is suggested to be “The Glycocalyx: Sugar Coating of Blood Vasculature in the Central Nervous System” 

Response: The title of the manuscript has been adjusted to highlight the topic of the GCX at the BBB rather than the entire CNS. 

“The Glycocalyx: Importance of Sugar Coating the Blood Brain Barrier” 

  1. In Ln 42-44, it was said “The GCX also includes proteins in the cell adhesion molecule family, such as the intercellular- 43 (ICAM), platelet endothelial- (PECAM), and vascular cell- (VCAM) adhesion molecules”, not sure if this statement is correct. Cell adhesion molecules are not components of the GCX. 

Response: Thank you for this comment. The sentence has been modified slightly to more accurately capture the biology of the glycocalyx and a reference has been added to this observation. 

“The GCX houses proteins in the cell adhesion molecule family, such as the intercellular- (ICAM), platelet endothelial- (PECAM), and vascular cell- (VCAM) adhesion molecules, shielding them from casual interaction”

  1. More detailed captions should be given for Fig. 1. 

Response: The caption for figure 1 has been updated to more completely describe the cartoon of the GCX.

Figure 1. “Representative comparison of the vascular glycocalyx in the periphery (a) and lining the interior cerebrovasculature at the BBB. (a) The mesh-layer of polysaccharides within a normal endothelial GCX includes from left to right glycolipids, hyaluronan with aggrecan containing brushes, glycoproteins, heparan sulfate (blue and purple chain) and chondroitin sulfate (solid blue chain) attached to syndecans, and glypicans [1]. (b) The endothelial GCX at the BBB contains the same major components as the peripheral GCX but exhibits a higher density, with notably higher levels of chondroitin sulfate and heparan sulfate expression, these function to stabilize and enmesh GCX components [8–10].”

  1. Ln 111-117, the length should be thickness. In addition, the observed thickness and coverage is from TEM. Sample preparations for different organs may have different artifacts for GCX coverage and thickness. This should be mentioned. 

Response: The authors appreciate the reviewer for pointing this out, this has been corrected. A sentence describing the importance of sample preparation lines 124-6: 

“It is important to note that GCX thickness and coverage in this study was observed using TEM, variation sample preparations may result in different observed thicknesses, however the trend remains consistent.” 

  1. It is suggested to have a list of acronyms for MACE-seq, LOC, HA, GAGs, KLF2, HAS2, t-MCAO, NO, Src, CM, etc. 

Response: Thank you for this suggestion, a complete table of acronyms can now be found at the end of the document as table 5. 

  1. Be consistent with the words used such as “syndecan-1” and “Syndecan1”. 

Response: Authors corrected wording to “syndecan-1” throughout. Authors looked through document to ensure that there were no other similar issues.

  1. There is no Fig. 3, “Fig. 3d” in Ln 615, etc, should be revised. Also other typos in the manuscript. 

Response: Authors revised “Fig. 3d” to “Fig. 2d” and addressed other typos in the manuscript. 

  1. Some recent articles were missing for the GCX of BBB, e.g.,  

Li et al. 2024 Cells 2024, 13, 190. https://doi.org/10.3390/cells13020190 

Xia et al, 2022 APL Bioeng. 6, 016101, 2022 https://doi.org/10.1063/5.0064381

Li et al, 2021 Cells 202110, 2576. https://doi.org/10.3390/cells10102576 

Li et al, 2023 Cells 2023, 12, 422. https://doi.org/10.3390/cells12030422 

It is suggested to perform more thorough literature review by using more combinations of keywords.

Response: The authors appreciate the suggestion to cite these papers from the Dr. Fu’s lab. We have added these citations, including an expanded section on DiGeorge Syndrome. We have also added a few additional citations using different sets of search terms.

Reviewer 3 Report

Comments and Suggestions for Authors

The endothelial glycocalyx (GCX) is a carbohydrate-rich layer that lines the luminal surface of endothelial cells throughout the vasculature, GCX and the blood-brain barrier (BBB) are critical components of vascular health, particularly in the central nervous system (CNS). Both structures play essential roles in maintaining homeostasis, protecting neural tissue, and regulating the exchange of substances between the blood and the brain. Their integrity is vital for the proper functioning of the brain and the prevention of neurological diseases. This manuscript summarized the current progression and knowledge of GCX in disease and how targeting the GCX at the BBB specifically for brain specific targeting for therapeutics, this information increased our knowledge about the mechanisms of GCX and CNS health, the targeted therapies hold promise for improving the therapeutics for CNS diseases. However, there are several major and minor weaknesses in the rationale and research methods of this work. Below please find the review comments. 

(1) Major comments

1. The major concern about this review manuscript is the topic is not focused, based on the title and keywords, the main topic should be GCX, BBB, and CNS (related diseases), as well as line 46 (This review will focus specifically on the role and importance of the GCX at the BBB.). However, the author also mentioned kidney disease (line253) and cancer (line 269-278), it is true that endothelial glycocalyx is involved in a couple of diseases, but I am feeling kind of lost during the reading, it is NOT necessary to mention all of the GCX-related disease, just focus on the CNS disease and BBB dysfunction. As I can find that Prof. Moriah’s research mainly focuses the human BBB in both health and disease, and she is an excellent expert in the BBB changes during stroke. I think more information and latest knowledge about BBB and stroke (mainly ischemic, hemorrhagic?) in the main text should be added. Moreover, the author mentioned CNS disease (and AD) for a couple of times without to many details, my suggestion is focus on stroke as priority and add AD (not generous CNS diseases) as 2nd point. Add a list or table if necessary to summarize the correlation of GCX, BBB and diseases.

2, Based on my understanding, the GCX and BBB integrity undergo a fluctuation of biphasic opening after ischemic stroke. It is essential to reflect this point in one of the sections, emphasize the biphasic change and the endothelial GCX, as well as the therapeutic strategies involved. Meanwhile, related figure(s) are required to describe this section.

3. It appears that the author tries to compare the normal GCX and BBB GCX in figure 1, however, I do not know how to define the normal GCX (non-endothelial GCX? Or non-BBB endothelial GCX?) and BBB GCX, as the author mentioned, the BBB is a complex network of endothelial cells, astrocytes, and pericytes. The author should clarify this point.

4. Page 13, line 615 and line 621 mentioned figure 3d, but I did not find the information about figure 3, please double check.

5. GCX is a complex network of glycoproteins, proteoglycans, and glycosaminoglycans. It is also far from understanding in various CNS diseases. It should be super cautious to talk about the therapeutic approaches, figure 2 should be modified to supply the potential therapeutic modulations of the GCX.

(2) Minor comments

1. Abbreviations should be properly defined and interpreted when they are first introduced in the manuscript. This practice ensures that readers can understand the meaning of abbreviations without confusion. It is also recommended to include a comprehensive list of abbreviations after the conclusion section. This list will serve as a quick reference for readers, enabling them to easily access the definitions throughout the manuscript.

2. When the author mentioned GCX within this manuscript, I believe it means the GCX of endothelial rather than other types of cell, this point should be emphasized, as GCX is not specific to endothelial. The author aims to use endothelial when talking about BBB and CNS diseases. Line 30 should be revised to specify the endothelial GCX.

3. Page 7, line 303, please specify the CNS disease, or add a list of these CNS diseases and BBB dysfunction.

Comments on the Quality of English Language

NA

Author Response

Reviewer 3

  1. The major concern about this review manuscript is the topic is not focused, based on the title and keywords, the main topic should be GCX, BBB, and CNS (related diseases), as well as line 46 (This review will focus specifically on the role and importance of the GCX at the BBB.). However, the author also mentioned kidney disease (line253) and cancer (line 269-278), it is true that endothelial glycocalyx is involved in a couple of diseases, but I am feeling kind of lost during the reading, it is NOT necessary to mention all of the GCX-related disease, just focus on the CNS disease and BBB dysfunction. As I can find that Prof. Moriah’s research mainly focuses the human BBB in both health and disease, and she is an excellent expert in the BBB changes during stroke. I think more information and latest knowledge about BBB and stroke (mainly ischemic, hemorrhagic?) in the main text should be added. Moreover, the author mentioned CNS disease (and AD) for a couple of times without to many details, my suggestion is focus on stroke as priority and add AD (not generous CNS diseases) as 2nd point. Add a list or table if necessary to summarize the correlation of GCX, BBB and diseases.

Response: The authors appreciate the point of this reviewer. It was important for us to include a selection of key results from other diseases and organ systems as these have helped us understand the GCX and the BBB. The GCX at the CNS is still vastly understudied and our understanding of how it contributes to CNS diseases such as ischemic stroke and AD are very limited. We have attempted to highlight this point to make it more apparent to the reader that there are insufficient studies in the CNS alone to make concrete conclusions and hypothesis about the GCX function.

We have also added tables to summarize the specific disease models of the GCX at the BBB table 1, and the wide array of model systems used tables 2-4 to help streamline the manuscript. 

  1. Based on my understanding, the GCX and BBB integrity undergo a fluctuation of biphasic opening after ischemic stroke. It is essential to reflect this point in one of the sections, emphasize the biphasic change and the endothelial GCX, as well as the therapeutic strategies involved. Meanwhile, related figure(s) are required to describe this section.

Response: We currently discuss the biphasic opening of the BBB and GCX in two locations. Clarification was made by authors in both sections to address this comment. Most of our understanding of changes at the GCX in ischemic stroke come from clinical observations and increased soluble plasma concentrations of GCX components. 

Included here is one of the major regions where this was discussed:

“It is well known that during an ischemic stroke there is a biphasic opening of the BBB, and this relationship is maintained when looking at the GCX breakdown [42]. Initially, the breakdown of the BBB during stroke and the subsequent reperfusion injury exacerbates endothelial GCX degradation resulting in increased inflammation and oxidative stress within the affected brain tissue [58]. One recent study observed this biphasic change pattern in the endothelial GCX during the first week following t-MCAO in mice, which corresponded to the biphasic evolution of permeability at the BBB [42]. In this study, the first phase following stroke resembled the degradation of the GCX caused by reperfusion injury. In contrast, the second phase corresponded to a restorative process with a recovered thickness of the GCX. Suggesting that the GCX at the BBB may transition from the dense representation in Figure 1b to the more closely mimic the more sparse GCX seen elsewhere in the body represented in Figure 1a. Specifically, levels of hyaluronan and syndecan-1 in plasma peaked at 6 hours following t-MCAO in mice, followed by a secondary peak after 7 days [42]. This observation acts as an indication of the potential of the GCX to reconstruct and repair itself following an ischemic event.

While these clear observations indicate that there is some change in the GCX following ischemic stroke, it is not entirely clear if this is entirely representative of the change on GCX composition in the brain or rather throughout the entire vascular system. A better understanding of this temporal relationship is needed. Understanding the dynamic nature of the endothelial GCX and the exploration of therapeutic strategies, such as biomimetic treatments, aimed at enhancing GCX repair and protecting BBB integrity after CNS injuries may be a valuable area for further study.“

  1. It appears that the author tries to compare the normal GCX and BBB GCX in figure 1, however, I do not know how to define the normal GCX (non-endothelial GCX? Or non-BBB endothelial GCX?) and BBB GCX, as the author mentioned, the BBB is a complex network of endothelial cells, astrocytes, and pericytes. The author should clarify this point. 

Response: The caption for figure 1 has been modified to more accurately describe the subject of the figure. 

“Representative comparison of the vascular glycocalyx in the periphery (a) and lining the interior cerebrovasculature at the BBB. (a) The mesh-layer of polysaccharides within a normal endothelial GCX includes from left to right glycolipids, hyaluronan with aggrecan containing brushes, glycoproteins, heparan sulfate (blue and purple chain) and chondroitin sulfate (solid blue chain) attached to syndecans, and glypicans [1]. (b) The endothelial GCX at the BBB contains the same major components as the peripheral GCX but exhibits a higher density, with notably higher levels of chondroitin sulfate and heparan sulfate expression, these function to stabilize and enmesh GCX components [8–10].”

  1. Page 13, line 615 and line 621 mentioned figure 3d, but I did not find the information about figure 3, please double check. 

Response: Changed to figure 2d.

  1. GCX is a complex network of glycoproteins, proteoglycans, and glycosaminoglycans. It is also far from understanding in various CNS diseases. It should be super cautious to talk about the therapeutic approaches, figure 2 should be modified to supply the potential therapeutic modulations of the GCX. 

Response: Figure 2 shows currently used methods of utilizing the GCX for therapeutic treatment. The caption for figure 2 has been modified to more clearly describe this. Figure 2: Potential avenues for therapeutic modulation of the GCX currently employed in preclinical studies. (a) The normal GCX includes hyaluronan, glycolipids, glycoproteins, heparan sulfate, chondroitin sulfate, syndecans, and glypicans [1]. (b) Sialic acid analogs and HDAC inhibitors enhance GCX GD2 expression in tumor cells. Improving anti-GD2 mAB targeting [120,121]. (c) Synthetic heparan sulfate modification of GCX attenuates Wnt signaling, boosting glucose clearance capacity [122]. (d) Antithrombin administration prevents endothelial GCX shedding during ischemia by stabilizing the GCX and promoting PGI2 release, inhibiting TNF-α-mediated degradation [123,124]. (e) Transient GCX disruption by tDCS increases BBB permeability, particularly for large or charged solutes [125].

(2) Minor comments 

  1. Abbreviations should be properly defined and interpreted when they are first introduced in the manuscript. This practice ensures that readers can understand the meaning of abbreviations without confusion. It is also recommended to include a comprehensive list of abbreviations after the conclusion section. This list will serve as a quick reference for readers, enabling them to easily access the definitions throughout the manuscript. 

Response: Thank you for this suggestion, table 5 has been added to the conclusion of the manuscript which includes all abbreviations used. We  have also ensured that each has been clearly defined in the text. 

  1. When the author mentioned GCX within this manuscript, I believe it means the GCX of endothelial rather than other types of cell, this point should be emphasized, as GCX is not specific to endothelial. The author aims to use endothelial when talking about BBB and CNS diseases. Line 30 should be revised to specify the endothelial GCX. 

Response: Thank you for this excellent clarification point. The authors have ensured that throughout the abstract and introduction it is more apparent that the topic of the review is specifically the endothelial glycocalyx. 

  1. Page 7, line 303, please specify the CNS disease, or add a list of these CNS diseases and BBB dysfunction. 

Response: To clarify this point, the reference for this particular sentence was focused on ischemic stroke and this information has been made more clear. Additionally Table 1 has been added to more clearly describe how the GCX is known to impact various CNS diseases.

Round 2

Reviewer 2 Report

Comments and Suggestions for Authors

This revision has addressed the previous concerns.